# The prevalence of mental illness in refugees and asylum seekers: A systematic review and meta-analysis

Rebecca Blackmore[1], Jacqueline A. Boyle[1], Mina Fazel[2], Sanjeeva Ranasinha[1], Kylie M. Gray[3,4], Grace Fitzgerald[1], Marie Misso[1], Melanie Gibson-Helm[1] *

**1** Monash Centre for Health Research and Implementation, School of Public Health and Preventive Medicine, Monash University, Melbourne, Australia, **2** Department of Psychiatry, Oxford University, Oxford, United Kingdom, **3** Centre for Developmental Psychiatry & Psychology, Department of Psychiatry, School of Clinical Sciences, Monash University, Melbourne, Australia, **4** Centre for Educational Development, Appraisal and Research (CEDAR), University of Warwick, Coventry, United Kingdom

* melanie.gibson@monash.edu

**Data Availability Statement:** All relevant data are within the manuscript and its Supporting Information files.

## Abstract

### Background

Globally, the number of refugees and asylum seekers has reached record highs. Past research in refugee mental health has reported wide variation in mental illness prevalence data, partially attributable to methodological limitations. This systematic review aims to summarise the current body of evidence for the prevalence of mental illness in global refugee populations and overcome methodological limitations of individual studies.

### Methods and findings

A comprehensive search of electronic databases was undertaken from 1 January 2003 to 4 February 2020 (MEDLINE, MEDLINE In-Process, EBM Reviews, Embase, PsycINFO, CINAHL, PILOTS, Web of Science). Quantitative studies were included if diagnosis of mental illness involved a clinical interview and use of a validated assessment measure and reported at least 50 participants. Study quality was assessed using a descriptive approach based on a template according to study design (modified Newcastle-Ottawa Scale). Random-effects models, based on inverse variance weights, were conducted. Subgroup analyses were performed for sex, sample size, displacement duration, visa status, country of origin, current residence, type of interview (interpreter-assisted or native language), and diagnostic measure. The systematic review was registered with PROSPERO (CRD) 42016046349. The search yielded a result of 21,842 records. Twenty-six studies, which included one randomised controlled trial and 25 observational studies, provided results for 5,143 adult refugees and asylum seekers. Studies were undertaken across 15 countries: Australia (652 refugees), Austria (150), China (65), Germany (1,104), Italy (297), Lebanon (646), Nepal (574), Norway (64), South Korea (200), Sweden (86), Switzerland (164), Turkey (238), Uganda (77), United Kingdom (420), and the United States of America (406). The prevalence of posttraumatic stress disorder (PTSD) was 31.46% (95% CI 24.43–38.5), the

**Funding:** The authors received no specific funding for this work. MG-H and JAB are supported by fellowships from the National Health and Medical Research Council. RB is supported by scholarships from Australian Rotary Health, Windermere Foundation, and Monash Centre for Health Research and Implementation (MCHRI).

**Competing interests:** The authors have declared that no competing interests exist.

**Abbreviations:** ASA-SI, Adult Separation Anxiety Semistructured Interview; CAPS, Clinician Administered PTSD Scale; CI, confidence interval; DSM, Diagnostic and Statistical Manual of Mental Disorders; ICD, International Classification of Disease; IS, Islamic State; M.I.N.I., the Mini-International Neuropsychiatric Interview; NA, not assessed in study; NOS, Newcastle-Ottawa Scale; NR, not reported; PTSD, posttraumatic stress disorder; RCT, randomised controlled trial; SCID, Structured Clinical Interview for DSM; SD, standard deviation; WHO, World Health Organization; WMH-CIDI, World Mental Health Composite International Diagnostic Interview.

prevalence of depression was 31.5% (95% CI 22.64–40.38), the prevalence of anxiety disorders was 11% (95% CI 6.75–15.43), and the prevalence of psychosis was 1.51% (95% CI 0.63–2.40). A limitation of the study is that substantial heterogeneity was present in the prevalence estimates of PTSD, depression, and anxiety, and limited covariates were reported in the included studies.

## Conclusions

This comprehensive review generates current prevalence estimates for not only PTSD but also depression, anxiety, and psychosis. Refugees and asylum seekers have high and persistent rates of PTSD and depression, and the results of this review highlight the need for ongoing, long-term mental health care beyond the initial period of resettlement.

## Author summary

### Why was this study done?

- Globally, the numbers of refugees and asylum seekers have reached record highs.

- This systematic review aims to estimate how common mental illnesses are in current adult refugee and asylum-seeker populations.

### What did the researchers do and find?

- We performed a comprehensive literature search looking for studies that diagnosed mental illness in refugee and asylum-seeker populations.

- For studies to be included, the diagnosis must have resulted from a clinical interview using a validated diagnostic assessment measure.

- We found adult refugee and asylum seekers have high and persistent rates of posttraumatic stress disorder (PTSD) and depression. The prevalence of anxiety disorders and psychosis are more comparable to findings from general populations.

### What do these findings mean?

- The increased prevalence of PTSD and depression appears to persist for many years after displacement.

- These results highlight the importance of early and ongoing mental health care, extending beyond the period of initial resettlement, to promote the health of refugees and asylum seekers.

## Introduction

Globally, the numbers of refugees and asylum seekers have reached record highs [1]. Ongoing conflicts around the world raise challenging social, political, and humanitarian issues [2]. For host-country health systems, the refugee crisis can have major implications for service planning and provision. Refugees and asylum seekers may have been exposed to traumatic events such as conflict, loss or separation from family, a life-threatening journey to safety, long waiting periods, and complexities with acculturation [3,4]. A sizable proportion are therefore at risk of developing psychological symptoms and major mental illness that can persist for many years after resettlement [5].

Estimates of the prevalence of mental illness in refugees vary greatly, even at the level of systematic reviews. Fazel and colleagues (2005) [6] conducted a systematic review and meta-analysis of refugees resettled in high-income countries, covering the period 1986–2004, and reported a prevalence of 9% for posttraumatic stress disorder (PTSD), 5% for major depressive disorder, and 4% for generalised anxiety disorder, based on studies reporting at least 200 participants. A subsequent systematic review into the association between torture or other traumatic events and PTSD and depression, covering studies between 1987 and 2009 and comprising 81,866 refugees and conflict-affected populations, reported an unadjusted weighted prevalence of 30% for PTSD and 30% for depression [7]. A recent systematic review of 8,176 Syrian refugees resettled in 10 countries reported a prevalence of 43% for PTSD, 40% for depression, and 26% for anxiety [8]. As the literature has focused on either specific cultural groups or specific host nations or has combined internally displaced populations with refugees and asylum seekers, there is a lack of estimates on the prevalence of mental illness in more representative refugee and asylum-seeker populations [9–12]. There is also a lack of research investigating the full breadth of mental illness, as the literature has mainly focused on PTSD and depression, hence the need for a comprehensive, worldwide, systematic review to investigate mental illness in the current refugee populations.

Some of the variation across individual studies may be attributable to methodological differences. For example, self-report measures tend to overestimate symptomatology, yet the literature relies heavily on these data rather than comprehensive psychiatric assessments using validated diagnostic tools [7,13]. There is also no uniform refugee experience: country of origin or resettlement, duration of displacement, or experience of displacement, amongst other important factors.

Given the changing nature of forced displacement and record numbers of refugees and asylum seekers, it is timely to reexamine this topic based on the many studies published since the two previously mentioned major reviews. Current prevalence information could be a powerful tool for advocacy and also assist host countries and humanitarian agencies to strengthen health services to provide the essential components of timely diagnosis and treatment for mental illnesses, in line with the priorities and objectives of the World Health Organization (WHO) Draft Global Action Plan 'Promoting the health of refugees and migrants' (2019–2023) [14]. Providing appropriate, early, and ongoing mental health care to refugees and asylum seekers benefits not only the individual but the host nation, as it improves the chances of successful reintegration, which has long-term benefits for the social and economic capital of that country, which will likely impact not only the displaced generation but the second generation as well [15]. Bringing together the global literature on the prevalence of mental illness in refugee and asylum-seeker populations would also enable the research community to move ahead and focus on different components of the mental health needs of this population, for example, on interventions, on less well-understood mental health conditions, or longitudinal mental health trajectories.

This systematic review aims to establish the current overall prevalence of mental illnesses in refugee and asylum-seeker populations by summarising the current global body of evidence and overcoming some methodological limitations of individual studies.

## Methods

### Search strategy and selection criteria

We followed the Preferred Reporting Items for Systematic Reviews and Meta-Analyses statement (S1 Prisma Checklist) [16] and registered the protocol with PROSPERO (record CRD42016046349) (https://www.crd.york.ac.uk/prospero/display_record.php?RecordID=46349). The search was based on that used in the earlier systematic review by Fazel and colleagues [6] but expanded to increase the range of databases searched, number of search terms, and stricter criteria regarding study inclusion. This review also placed no restrictions on resettlement countries. In total, eight databases were searched: MEDLINE, MEDLINE In-Process, EBM Reviews, Embase, PsycINFO, CINAHL, PILOTS, Web of Science. The search strategy included terms for refugees and asylum seekers and terms related to mental illness, diagnosis, and trauma. An example of a complete search string is provided in S1 Table. The date limits of the search were 1 January 2003 to 4 February 2020. This start date reflects the end date of the search conducted by Fazel and colleagues [6], in order to provide a contemporary estimate of mental illness within this population. The reference lists of 92 relevant systematic reviews identified during the search were also screened, resulting in an additional 37 articles to review.

Studies were included if (1) the sample solely comprised adult refugees and/or asylum seekers residing outside their country of origin, (2) had a sample size larger than 50, and (3) reported quantitative prevalence estimates of a mental illness as classified by the Diagnostic and Statistical Manual of Mental Disorders (DSM) [17] or the International Classification of Disease (ICD) [18]. This diagnosis must have resulted from a clinical interview using a validated diagnostic assessment measure. The interview needed to be conducted either by a mental health professional (psychiatrist, psychologist, psychiatric nurse) or trained paraprofessional (psychology research assistant, trained researcher). In studies administering the WHO World Mental Health Composite International Diagnostic Interview (WMH-CIDI) [19], nonclinicians who had completed official WHO training were accepted, as this fully structured interview measure is intended for use by trained lay interviewers. If multiple articles reported data from the same study, the article providing data best meeting the selection criteria was included. Randomised controlled trials (RCTs), longitudinal cohort, and cross-sectional studies were considered for inclusion, whereas retrospective registry reviews, medical records audits, and qualitative studies were excluded. Case-control studies were excluded if cases were selected based on the presence of our outcomes of interest.

Studies were excluded if they met the following criteria:

- Participants were recruited from psychiatric or mental health clinics (to reduce selection bias). However, those studies that recruited participants from primary healthcare clinics were still included.

- The sample included asylum seekers whose applications had been rejected and the results were not disaggregated or the assessment was not conducted prior to rejection (when the individuals met the definition of asylum seekers).

- Diagnoses were based solely on self-report questionnaires or symptom rating scales.

Two reviewers (RB and MG-H or GF) independently assessed the title, abstract, and keywords of every article retrieved against the selection criteria. Full text was then assessed if the

title and abstract suggested the study met the selection criteria. We contacted 31 study authors for further information regarding methodology and data and received 28 responses. Studies in languages other than English were assessed first by a native speaker when possible or via Google translate and then professionally translated if assessed as potentially eligible.

## Data analysis

Using a fixed protocol, two review authors (RB and MG-H) independently extracted statistical data and study characteristics: host country, publication year, sample size, country or region of origin, sampling method, diagnostic tool and criteria, use of interpreter, age, proportion of female participants, visa status, duration of displacement, and prevalence of mental illness (numerator and denominator). Data regarding the sex distribution of samples were extracted separately for males and females, when possible.

Meta-analysis results (Stata software version 14.1 [StataCorp]) were expressed as prevalence estimates of mental illness calculated with 95% confidence intervals (CIs) in the pooled data. Random-effects meta-analyses using a DerSimonian and Laird estimator based on inverse variance weights were employed [20]. Random-effects meta-analysis was chosen, as heterogeneity was anticipated because of between-study variations in clinical factors due to the heterogenous nature of refugees and asylum seekers (e.g., country of origin, language, host nations, etc.). The DerSimonian and Laird method incorporates a measure of the heterogeneity between studies. Heterogeneity was assessed using the $I^2$ statistic [21]. In the case of five or more studies being available, publication bias was assessed by visual inspection of funnel plots and applying Egger's test set at a threshold of a $p$-value less than 0.05 to indicate funnel plot asymmetry [22]. Prevalence rates were for current diagnoses, except studies reporting 1-year prevalence as assessed by the WHO WMH-CIDI [23–25].

Sources of heterogeneity between studies were investigated, when reported data allowed, by subgroup analyses. This included sex, sample size, displacement duration, visa status, country or region of origin, current residence, type of interview (interpreter-assisted or native language), and diagnostic measure. As prevalence of mental illness is related to sample size [6], the subgroup analysis for sample size compared studies with more or less than 200 participants.

## Risk of bias assessment

Methodological quality was independently assessed by two reviewers (RB and JAB) using an assessment template for risk of bias, developed a priori according to study design, which meant the criteria to assess an RCT were different from the criteria of an observational study (S1 Risk of Bias) [26]. These templates are based upon the Newcastle-Ottawa Scale (NOS) [27], with the addition of further risk of bias components assessing internal and external validity such as use of appropriate study design, explicit and appropriate use of inclusion criteria, reporting bias, confounding, sufficient power for analyses, and any apparent conflicts of interest, as has been used in international evidence-based guidelines and other systematic reviews [28–30]. Using a descriptive approach, studies were assigned a rating of low, moderate, or high risk of bias. Any disagreement was resolved by discussion with other reviewers (MG-H and MF) to reach a consensus. Such discussions occurred on two occasions, both times regarding papers assigned at high risk of bias [31,32].

## Results

The entire search yielded 21,842 records (Fig 1). After removing duplicates, 12,517 records were excluded based on title and abstract and a further 1,186 records were selected for full text

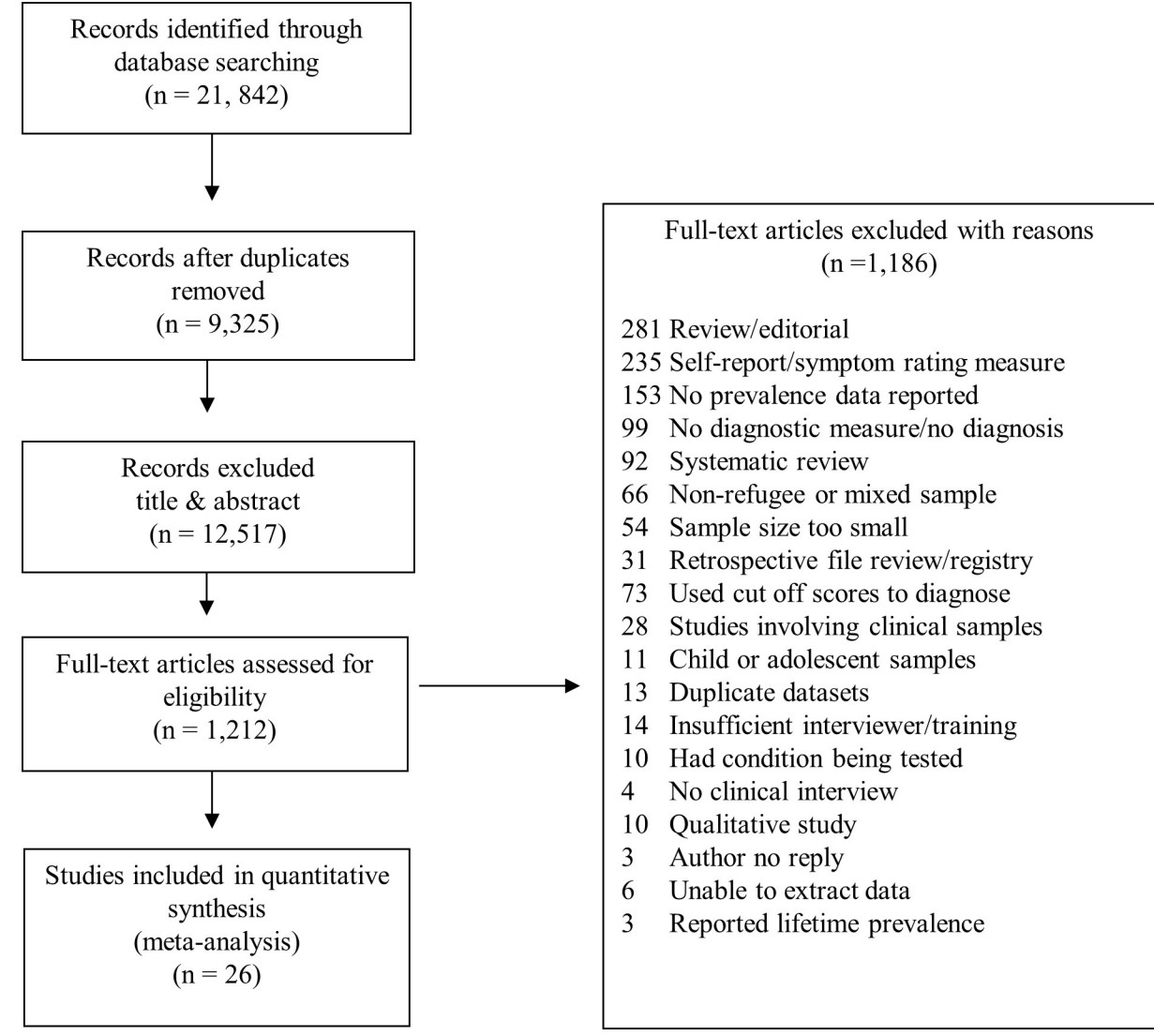

**Fig 1. Search results and selection of studies reporting prevalence of mental illness among refugees and asylum seekers.**

review. Twenty-six studies met the inclusion criteria, providing data for 5,143 adult refugees and asylum seekers (Fig 1). Characteristics of the included studies are provided in Table 1. All were observational, except one RCT from which we included baseline prevalence data [24]. Studies were undertaken in 15 countries: Australia (652 refugees) [33–37], Austria (150) [38], China (65) [32], Germany (1,104) [39–44], Italy (297) [39], Lebanon (646) [45,46], Nepal (574) [25], Norway (64) [23], South Korea (200) [47], Sweden (86) [48], Switzerland (164) [49,50], Turkey (238) [51], Uganda (77) [24], UK (420) [39,52], and the US (406) [31,53]. Participants were from four geographic regions: the Middle East (43%), Europe (29%), Asia (20%), and Africa (5%), with two studies reporting refugee samples coming from 18 different countries (3%) [26, 41] (97% of total sample due to unreported countries of origin).

Five diagnostic measures were used (S1 Table): Structured Clinical Interview for DSM (SCID) [54], Mini-International Neuropsychiatric Interview (M.I.N.I.) [55], Clinician Administered PTSD Scale (CAPS) [56], and WHO WMH-CIDI [18]. None of these instruments were developed specifically for refugee populations but have been widely used in different cultural

**Table 1. Characteristics of included studies.**

| Study | Country or Region of Origin | Sampling | Instrument and Criteria | Interview in Native Language | N | Age: Years M (SD) | Female: % | PTSD (%) | DEP (%) | ANX (%) | PSY (%) | Risk of Bias |
|---|---|---|---|---|---|---|---|---|---|---|---|---|
| Bogic et al., 2012 [39] (Germany, Italy, and UK) | Former Yugoslavia | In Germany and Italy, refugees contacted via resident registry lists. In UK, refugees contacted via community organizations and snowball techniques. | M.I.N.I. DSM-IV | Yes | 854 | 41.6 (10.8) | 51.29 | 283 (33.14) | 292/ 851 (34.31) | 74/854 (7.36) | 11 (1.29) | Mod. |
| Charney and Keane, 2007 [31] (USA) | Former Yugoslavia | Advertised psychological treatment study for Bosnian refugees suffering the effects of the Balkans' civil war. | SCID DSM-IV | Yes | 115 | 46 (13.78) | 67 | NA | NA | NA | 9 (8) | High |
| Eckart et al., 2011 [40] (Germany) | Albania, Serbia, Romania, and Turkey | Recruited participants from shelters for asylum seekers and Kurdish recreational facilities. | CAPS and M.I.N.I. DSM-IV | No | 52 | PTSD group: 36.2 (7.7) Trauma controls: 34.1 (9.9) Nontrauma controls: 29 (7.2) | 0 | 20 (38.46) | 17 (32.69) | NA | NA | Low |
| Heeren et al., 2012 [49] (Switzerland) | Europe, Africa, and Asia | Two groups sampled consecutively from lists provided every 2 weeks for 6 months from the national register of adult asylum seekers in Switzerland. | M.I.N.I. DSM-IV | No | 86 | Group 1: 26.7 (7.2) Group 2: 32.9 (9.6) | 30.23 | 20 (23.25) | 27 (31.39) | 7 (8.13) | NA | Low |
| Hocking et al., 2018 [36] (Australia) | Africa and Asia | Consecutive sample populations of 'general-access-listed' clients at ASRC and Refugee Health Clinic, Dental Clinic in Victoria, Australia. | M.I.N.I. DSM-IV | No | 185 | 33 | 30.3 | 38 (20.7) | 56 (30.3) | 6 (3.2) | 2 (1) | Mod. |
| Jakobsen et al., 2011 [23] (Norway) | Middle East, North Africa, Somalia, and former Yugoslavia | 12 reception centres, all eligible asylum seekers (i.e., stay in Norway 4 months, age > 18 years, and speakers of one of the included languages). | WHO-CIDI DSM-IV | No | 64 | 33 (11.6) | 46.88 | 29 (4.31) | 21 (32.81) | 17 (26.56) | 1 (1.56) | Low |
| Jeon et al., 2005 [47] (South Korea) | North Korea | All North Korean refugees living in Seoul (July 1998–November 2000) were contacted via telephone and asked to participate. | SCID DSM-III | Yes | 200 | 34.7 (10.3) | 41.5 | 59 (29.50) | NA | NA | NA | Low |
| Kazour et al., 2017 [46] (Lebanon) | Syria | Household survey on refugees between 18 and 65 years old in six Central Bekaa camps in Lebanon. | M.I.N.I. DSM-IV | Yes | 452 | 35.05 (12.35) | 55.75 | 123 (27.21) | NA | NA | NA | Low |
| Kizilhan, 2018 [42] (Germany) | Iraq | Participants were part of special quota project in Baden-Wuerttemberg to support women escaped from IS. | SCID DSM-IV | Yes | 296 | 23.72 (2.6) | 100 | 144 (48.6) | 158 (53.4) | 116 (39.1) | NA | High |

*(Continued)*

**Table 1.** (Continued)

| Study | Country or Region of Origin | Sampling | Instrument and Criteria | Interview in Native Language | N | Age: Years M (SD) | Female: % | PTSD (%) | DEP (%) | ANX (%) | PSY (%) | Risk of Bias |
|---|---|---|---|---|---|---|---|---|---|---|---|---|
| Llosa et al., 2014 [45] (Lebanon) | Palestine | Selected households chosen from the Burj el-Barajneh camp in southern Beirut, Lebanon. | M.I.N.I. DSM-IV | Yes | 194 | 41.5 (15) | 71.13 | 9 (4.64) | 31 (15.98) | 15 (7.73) | 5 (2.58) | Low |
| Maier et al., 2010 [50] (Switzerland) | 18 different countries: Asia, Africa, and Europe | List provided by Swiss Federal Office for Migration, all adult (18 + years old) asylum seekers applying after 1 August 2007 and assigned to the Zurich canton. | M.I.N.I. DSM-IV | No | 78 | 29.9 (8.4) | 26.92 | 19 (24.36) | 26 (33.33) | 8 (10.26) | NA | Low |
| Momartin et al., 2004 [33] (Australia) | Former Yugoslavia | The Bosnian Resource Centre provided a list of names. In order to obtain additional participants, a snowball technique was also utilised. | CAPS and SCID DSM-IV | No | 126 | 47 (NR) | 61.11 | 79 (62.70) | 58 (46.03) | NA | NA | Mod. |
| Neuner et al., 2004 [24] (Uganda) | Sudan | Participants randomly chosen from list of respondents who had previously been randomly selected in a hut-to-hut survey at the Imvepi Settlement in Uganda. | WHO-CIDI DSM-IV | No | 77 | NR (NR) | NR | 43 (55.84) | NA | NA | NA | Low |
| Rees et al., 2019 [37] (Australia) | Middle East, Sri Lanka, and Sudan | The study was conducted at three public antenatal clinics in Sydney and Melbourne, Australia. At first appointment, women were identified by clinic records through requests for interpreters, culturally recognizable surname, and country of birth data. | M.I.N.I. DSM-IV | Yes | 289 | 30 (5.8) | 100 | NA | 94 (32.5) | NA | NA | Low |
| Renner et al., 2006 [38] (Austria) | Chechnya, West Africa, Afghanistan | All participants had applied for political asylum in Austria. | CAPS DSM-IV | No | 150 | Chechnya: 32.4 (10.7) West Africa: 32.5 (7.1) Afghanistan: 27.5 (9) | 26.67 | 38 (25.33) | NA | NA | NA | Low |
| Richter et al., 2018 [43] (Germany) | Middle East, Russia, Azerbaijan | Asylum seekers from an admission centre in southern Germany. Two samples; help seekers, those responding to flyers regarding psychiatric services, and random sample, randomly selected residents of the centre. | M.I.N.I. ICD-10 | Yes | 283 | 31.9 (10.6) | 44 | 58 (20.5) | 62 (21.9) | 11 (3.8) | 3 (1) | High |

(*Continued*)

**Table 1.** (Continued)

| Study | Country or Region of Origin | Sampling | Instrument and Criteria | Interview in Native Language | N | Age: Years M (SD) | Female: % | PTSD (%) | DEP (%) | ANX (%) | PSY (%) | Risk of Bias |
|---|---|---|---|---|---|---|---|---|---|---|---|---|
| Silove et al., 2010 [34] (Australia) Data obtained from same refugee population used by Momartin et al., 2004 [33] | Former Yugoslavia | The Bosnian Resource Centre provided a list of names. In order to obtain additional participants, a snowball technique was also utilised. | ASA-SI DSM-IV | No | 126 | 47 (NR) | 61.11 | NA | NA | 22 (17.46) | NA | Mod. |
| Sondergaard and Theorell, 2004 [48] (Sweden) | Iraq | Recently resettled refugees from Iraq. | CAPS DSM-IV | No | 86 | 34.7 (7.7) | 37.21 | 32 (37.21) | NA | NA | NA | Low |
| Tay et al., 2013 [35] (Australia) | Refugees in Australia from 18 different countries covering Middle East, Africa, Asia | Participants selected using cluster-probabilistic sampling method. Randomly approached 87 migration agents who had represented asylum seekers during a 12-month period (2001–2002). | SCID DSM-IV | No | 52 | 39 (13.5) | 34.61 | 31 (59.61) | 30 (57.69) | NA | NA | Low |
| Tekin et al., 2016 [51] (Turkey) | Iraq | Yazidi refugees displaced from Shengal region in Iraq and entered Turkey between July and September 2014 and living in camp (February–April 2015) in the Cizre district of Turkey. | SCID DSM-IV | Yes | 238 | 32.7 (11.87) | 55.88 | 102 (42.86) | 94 (39.50) | NA | NA | Mod. |
| Turner et al., 2003 [52] (UK) | Kosovo | Participants recruited from five reception centres in the north of England for refugees from Kosovo (November 1999–January 2000). | CAPS DSM-IV | Yes | 118 | 37.1 (14.7) | 53.33 | 46 (38.98) | NA | NA | NA | Mod. |
| Van Ommeren et al., 2004 [25] (Nepal) | Bhutan | Participants randomly selected from United Nations camp list of Bhutanese refugees. | WHO-CIDI ICD-10 | Yes | 574 | Shamans: 51.3 (11.7) Nonhealers: 43.7 (12.9) | 0 | 154 (26.8) | 11 (1.92) | 27 (4.70) | NA | Mod. |
| von Lersner et al., 2008 [41] (Germany) | Bosnia, Serbia, Kosovo, Iraq, Turkey | Participants recruited by advertisements posted in refugee centres, language schools, and doctors' offices. Organizations involved in the return of refugees were contacted. | M.I.N.I. DSM-IV | No | 100 | 43.2 (14.9) | 50 | NA | 42 (42.00) | 2 (2.00) | NA | Mod. |
| Wright et al., 2017 [53] (USA) | Iraq | Adult Iraqi refugees randomly selected from population who arrived in Michigan between October 2011 and August 2012. Recruited with collaboration of three resettlement agencies. | SCID DSM-IV | Yes | 291 | 34.30 (11.37) | 45.7 | 11 (3.78) | 8 (2.75) | NA | NA | Low |

(Continued)

**Table 1.** (Continued)

| Study | Country or Region of Origin | Sampling | Instrument and Criteria | Interview in Native Language | N | Age: Years M (SD) | Female: % | PTSD (%) | DEP (%) | ANX (%) | PSY (%) | Risk of Bias |
|-------|------------------------------|----------|--------------------------|------------------------------|---|--------------------|-----------|----------|---------|---------|---------|--------------|
| Wulfes et al., 2019 [44] (Germany) | Middle East and Sudan | Asylum seekers living in refugee accommodation (Braunschweig) Residents were asked to participate by staff at centre, social workers, research team, and flyers. | SCID DSM-5 | No | 118 | 32.9 (13.1) | 35.6 | 35 (29.7) | 39 (33.1) | NA | NA | Mod. |
| Yu and Jeon, 2008 [32] (China) | North Korea | Refugees over 15 years of age who were in protective facilities in China under the South Korean government protection. | SCID DSM-IV | Yes | 65 | NR (NR) | 70.77 | 3 (4.61) | NA | NA | NA | High |

Abbreviations: ANX, anxiety; ASA-SI, Adult Separation Anxiety Semistructured Interview; ASRC, Asylum Seeker Resource Centre; CAPS, Clinician Administered PTSD Scale; DEP, depression; DSM-IV, Diagnostic and Statistical Manual of Mental Disorders, Fourth Edition; ICD-10, International Classification of Disease, 10th Edition; IS, Islamic State; M, mean; Mod., moderate; M.I.N.I., the Mini-International Neuropsychiatric Interview; N, number; NA, not assessed in study; NR, not reported PSY, psychosis; PTSD, posttraumatic stress disorder; SCID, Structured Clinical Interview for DSM; SD, standard deviation; WHO-CIDI, World Health Organization–Composite International Diagnostic Interview

contexts. Nine studies mentioned the reliability or validity of the used instruments [23,35,36,39,41,44–46,51]. Thirteen studies conducted the assessment in the refugee's native language [25,31,32,37,39,42,43,45–47,51–53]. Thirteen studies were conducted with assistance from interpreters [23,24,33–36,38,40,41,44,48–50].

Twenty-two studies of PTSD were identified ($n$ = 4,639) [23–25,32,33,35,36,38–40,42–53]. Participants had a weighted mean age of 35.2 years and 44% were women. Overall, 31.46% (95% CI 24.43–38.50) were diagnosed with PTSD (1,376/4,639) (Fig 2). There was substantial heterogeneity between studies (Fig 2), and subgroup analyses indicated PTSD prevalence was significantly higher for women (34.02%, 95% CI 31.12–37.01, $p$ = 0.02), in the smaller studies ($n$ < 200) (37.35%, 95% CI 34.86–39.90, $p$ < 0.001), those with refugee status (31.01%, 95% CI 29.52–32.54, $p$ < 0.001), and those originating from Africa (48.25%, 95% CI 39.82–56.75, $p$ < 0.001) (Fig 3). In the eight largest studies with 200 participants or more, PTSD prevalence was significantly lower (29.30%, 95% CI 27.72–30.91, $p$ < 0.001) [25,39,42,43,46,47,51,53]. Duration of displacement had no significant impact on PTSD prevalence ($p$ = 0.11). The prevalence of PTSD for those displaced less than 4 years was 30.17% (95% CI 28.24–32.14) compared to 33.14% (95% CI 29.99–36.41) for those displaced longer than 4 years. The PTSD prevalence for interpreter-assisted interviews was 35.75% (95% CI 33.80–39.70) compared to 27.82% (95% CI 26.40–29.30) for interviews conducted in the native language ($p$ < 0.001). There was a statistically significant difference across diagnostic measures ($p$ < 0.001) with the CAPS yielding a higher prevalence of PTSD (40.41%, 95% CI 36.20–44.70), followed by the WHO-CIDI (31.6%, 95% CI 28.20–35.20), the SCID (30.55%, 95% CI 28–33.20), and the M.I.N.I. (25.8%, 95% CI 24–27.70).

Seventeen studies of depression were identified ($n$ = 3,877) [23,25,33,35–37,39–45,49–51,53]. Participants had a weighted mean age of 35.7 years and 48% were women. Overall, 31.51% (95% CI 22.64–40.38) were diagnosed with depression (1,066/3,877) (Fig 4). Three studies provided separate data for dysthymia ($n$ = 1,135) [39,41,45]. The overall prevalence of dysthymia was 6.72% (95% CI 3.63%–9.81%) with moderate heterogeneity between studies

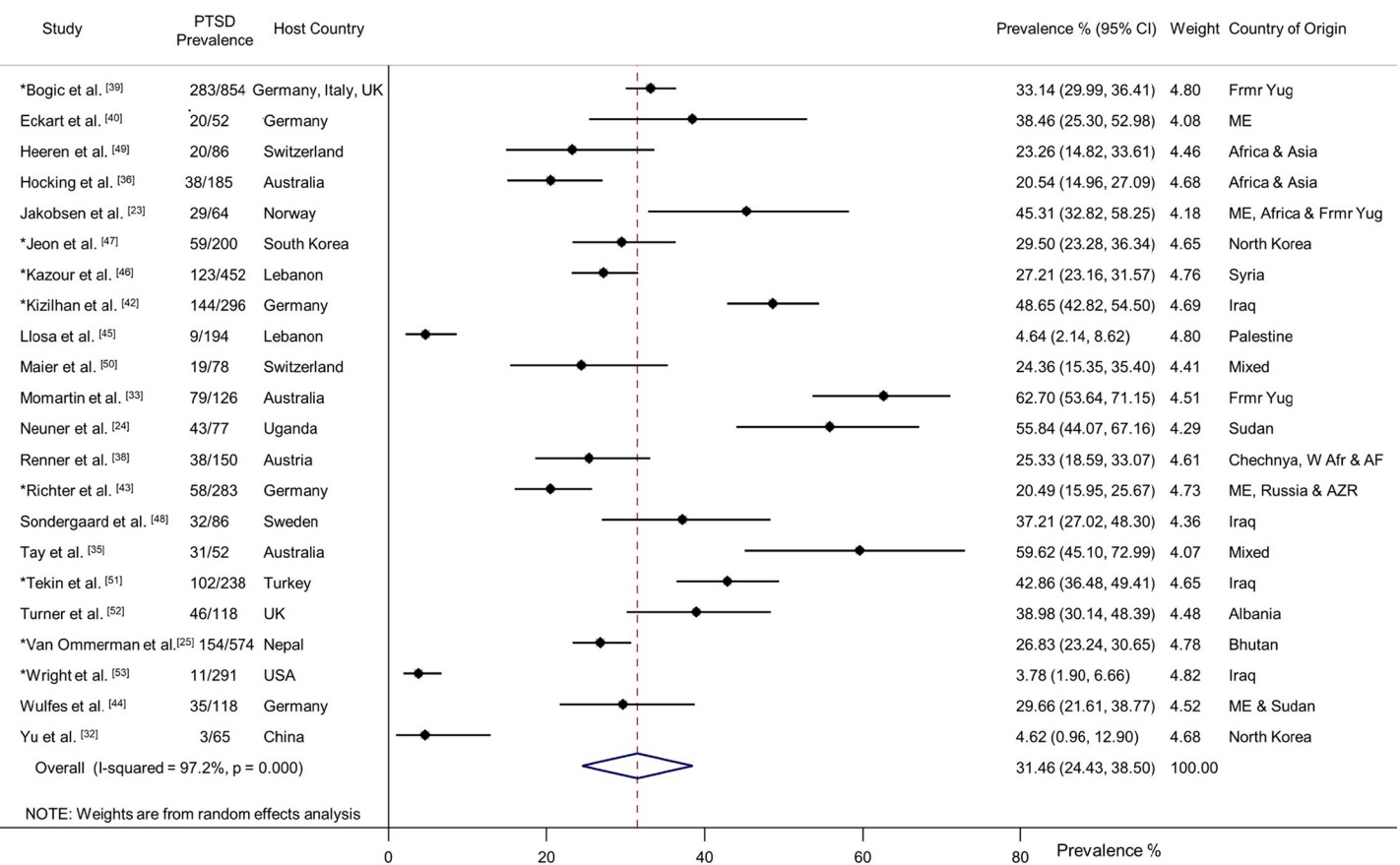

**Fig 2. Prevalence of PTSD in refugees and asylum seekers.** *Study with sample size of ≥200. Horizontal lines indicate 95% CIs; horizontal points of the open diamond are the limits of the overall 95% CIs; and the red dashed line shows the position of the overall prevalence. AF, Afghanistan; AZR, Azerbaijan; CI, confidence interval; Frmr Yug, former Yugoslavia; ME, Middle East; PTSD, posttraumatic stress disorder; W Afr, West Africa.

($I^2$ = 65.6%, $p$ = 0.055). There was considerable heterogeneity between the studies (Fig 4). Subgroup analyses indicated depression prevalence was significantly higher in the smaller studies 32.89% (95% CI 30.06–35.82, $p < 0.001$), for those deemed asylum seekers 30.14% (95% CI 27.10–33.32, $p = 0.04$), those originating from Europe 35.82% (95% CI 32.81–38.92, $p < 0.0001$), and for those living in the community 30.70% (95% CI 28.74–32.72, $p < 0.0001$) (Fig 5). The subgroup analysis for sex could not be conducted, owing to a lack of reported data. In the seven larger studies with 200 or more participants [25,37,39,42,43,51,53], the reported depression prevalence was 20.65% (95% CI 18.88–22.51), which was significantly lower ($p < 0.001$) than in the smaller studies, 32.89% (95% CI 30.06–35.82). Duration of displacement had no significant impact on depression prevalence ($p = 0.17$). The prevalence of depression for those displaced less than 4 years was 32.44% (95% CI 30.00–34.95) and 35.12% (95% CI 32.08–38.25) for those displaced longer than 4 years. The depression prevalence for interpreter-assisted interviews was 35.35% (95% CI 32.05–38.76) compared to 24.87% (95% CI 23.33–26.45) for interviews conducted in the native language ($p < 0.0001$). There was a statistically significant difference across type of diagnostic measures ($p < 0.0001$) with the SCID yielding a higher prevalence of depression (34.52%, 95% CI 31.74–37.39), followed by the M.I.N.I. (30.55%, 95% CI 28.59–32.56) and the WHO-CIDI (5.02%, 95% CI 3.46–7.01).

Eleven studies of anxiety disorders were identified ($n$ = 2,840) [23,25,34,36,39,41–43,45,49,50]. Participants had a weighted mean age of 36.8 years and 31% were women. Four

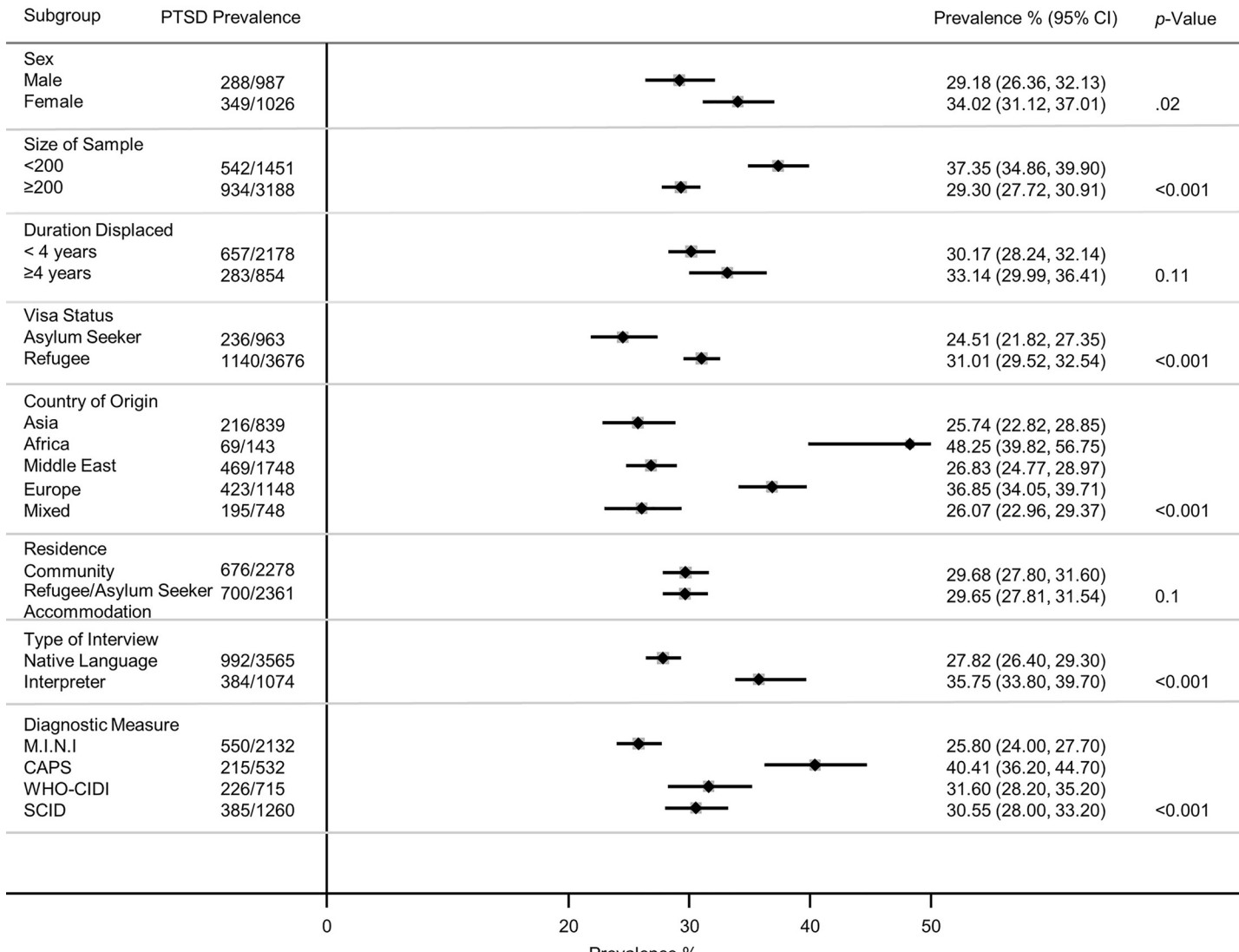

**Fig 3. Prevalence of PTSD by various study characteristics.** *p*-Values derived from random-effects models; horizontal lines indicate 95% CIs. CAPS, Clinician Administered PTSD Scale; CI, confidence interval; M.I.N.I., the Mini-International Neuropsychiatric Interview; PTSD, posttraumatic stress disorder; SCID, Structured Clinical Interview for DSM; WHO-CIDI, World Health Organization–Composite International Diagnostic Interview.

studies reported prevalence for generalised anxiety disorder [25,39,41,45], six reported any anxiety disorder [23,36,42,43,49,50], and one study diagnosed adult separation anxiety disorder [34]. Overall, 11.09% (95% CI 6.75–15.43) were diagnosed with an anxiety disorder (305/2,840) (Fig 6). There was substantial heterogeneity between studies (Fig 6). Subgroup analyses indicated anxiety prevalence was higher for those displaced less than 4 years (21.72%, 95% CI 18.74–24.94, *p* < 0.0001), those granted formal refugee status (11.44%, 95% CI 10.12–12.87, *p* = 0.0009), those originating from the Middle East (26.73%, 95% CI 22.86–30.89, *p* < 0.0001), and those living in temporary refugee accommodation (13.18%, 95% CI 11.46–15.06, *p* < 0.0001) (Fig 7). The subgroup analysis for sex could not be conducted, owing to a lack of reported data. Sample size had no significant impact on anxiety disorder prevalence (*p* = 0.21). The prevalence of anxiety disorders in the smaller studies (*N* < 200) was 9.24% (95% CI 7.36–11.42), and in the larger studies (*N* ≥ 200), the prevalence was 10.83% (95% CI 9.50–12.27).

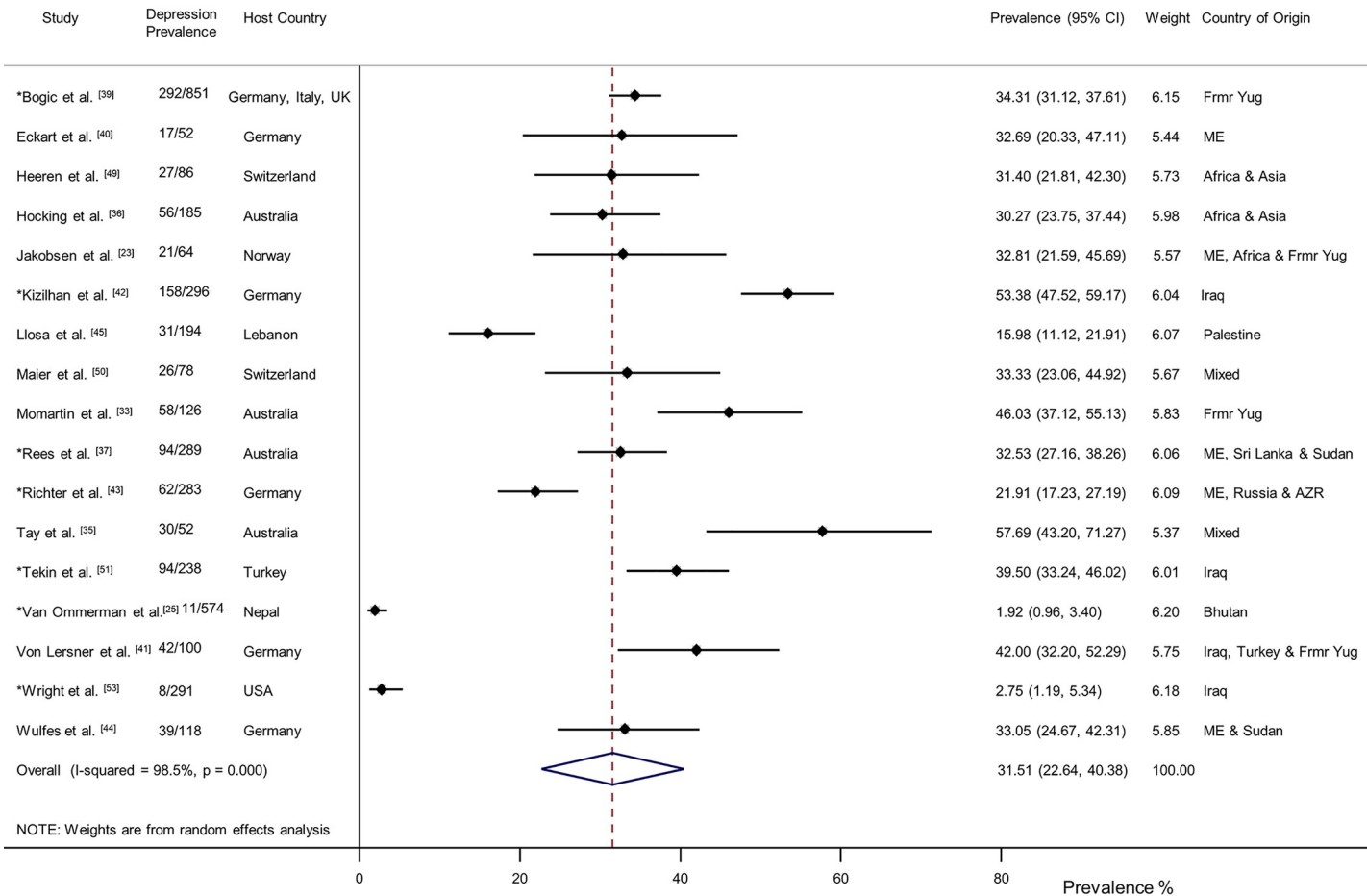

**Fig 4. Prevalence of depression in refugees and asylum seekers.** *Study with sample size of ≥200. Horizontal lines indicate 95% CIs; horizontal points of the open diamond are the limits of the overall 95% CIs; and the red dashed line shows the position of the overall prevalence. AZR, Azerbaijan; CI, confidence interval; Frmr Yug, former Yugoslavia; ME, Middle East.

The use of an interpreter to conduct assessments had no significant impact on the reported prevalence of anxiety disorders (*p* = 0.34). The prevalence of anxiety for interpreter-assisted interviews was 9.70 (95% CI 7.50–12.30) and 11.04% (95% CI 9.76–12.40) for those interviews conducted in the native language. The subgroup analysis for diagnostic measure could not be conducted, owing to insufficient studies for each measure.

Six studies of psychotic illness were identified (*n* = 1,695) [23,31,36,39,43,45]. Participants had a weighted mean age of 37.6 years and 51% were female. Overall, 1.51% (95% CI 0.63–2.40) were diagnosed with psychosis (31/1,695), with low heterogeneity between studies (Fig 8).

## Publication bias

There was no evidence of publication bias for PTSD, depression, anxiety, or psychosis (S1–S4 Egger's Tests).

## Risk of bias

Thirteen studies were assigned a low risk of bias and determined to be of high quality [23,24,35,37,38,40,45–50,53]. Nine studies demonstrated moderate risk of bias

| Subgroup | Depression Prevalence | | Prevalence % (95% CI) | p-Value |
|---|---|---|---|---|
| **Size of Sample** | | | | |
| < 200 | 347/1055 | | 32.89 (30.06, 35.82) | |
| ≥ 200 | 405/1961 | | 20.65 (18.88, 22.51) | <0.001 |
| **Duration Displaced** | | | | |
| < 4 years | 459/1415 | | 32.44 (30.00, 34.95) | |
| ≥ 4 years | 334/951 | | 35.12 (32.08, 38.25) | 0.17 |
| **Visa Status** | | | | |
| Asylum Seeker | 261/866 | | 30.14 (27.10, 33.32) | |
| Refugee | 805/3018 | | 26.67 (25.10, 28.29) | 0.04 |
| **Country of Origin** | | | | |
| Asia | 11/574 | | 1.92 (0.96, 3.40) | |
| Africa | 7/30 | | 23.33 (9.93, 42.28) | |
| Middle East | 308/1078 | | 28.57 (25.89, 31.37) | |
| Europe | 350/977 | | 35.82 (32.81, 38.92) | |
| Mixed | 275/866 | | 31.76 (28.66, 34.97) | <0.0001 |
| **Residence** | | | | |
| Community | 650/2117 | | 30.70 (28.74, 32.72) | |
| Refugee/Asylum Seeker Accommodation | 416/1767 | | 23.54 (21.58, 25.59) | <0.0001 |
| **Type of Interview** | | | | |
| Native Language | 750/3016 | | 24.87 (23.33, 26.45) | |
| Interpreter | 286/809 | | 35.35 (32.05, 38.76) | <0.0001 |
| **Diagnostic Measure** | | | | |
| M.I.N.I | 647/2118 | | 30.55 (28.59, 32.56) | |
| WHO-CIDI | 32/638 | | 5.02 (3.46, 7.01) | |
| SCID | 387/1121 | | 34.52 (31.74, 37.39) | <0.0001 |

0   20 30 40 50
Prevalence %

**Fig 5. Prevalence of depression by various study characteristics.** p-Values derived from random-effects models; horizontal lines indicate 95% CI. Subgroup analysis for sex could not be conducted, owing to a lack of reported data. CI, confidence interval; M.I.N.I., the Mini-International Neuropsychiatric Interview; SCID, Structured Clinical Interview for DSM; WHO-CIDI, World Health Organization–Composite International Diagnostic Interview.

[25,33,34,36,39,41,44,51,52]. A moderate rating was assigned to studies that had issues with the representativeness of their sample or used nonrandom sampling techniques. Additionally, in one study, only male psychologists conducted the diagnostic assessments, and this was associated with fewer than expected reports of sexual assault [51]. Four studies were assigned a high risk of bias [31,32,42,43]. One study, providing data for PTSD and depression, assessed the mental health consequences of captivity by the Islamic State (IS) militant group on a sample of Yazidi women. It was reported that some of the women were not yet ready to receive psychotherapy for their symptoms [42]. This may have impacted upon the reported prevalence rates, particularly PTSD, as some women may have been reluctant and not ready to disclose trauma details during the research interviews. Another study, providing PTSD data, conducted diagnostic assessments in nonconfidential areas of a detention facility [32]. The reported PTSD prevalence was low but similar to two other studies assigned a low risk of bias. Two studies recruited help-seeking populations through the use of advertisements or flyers offering

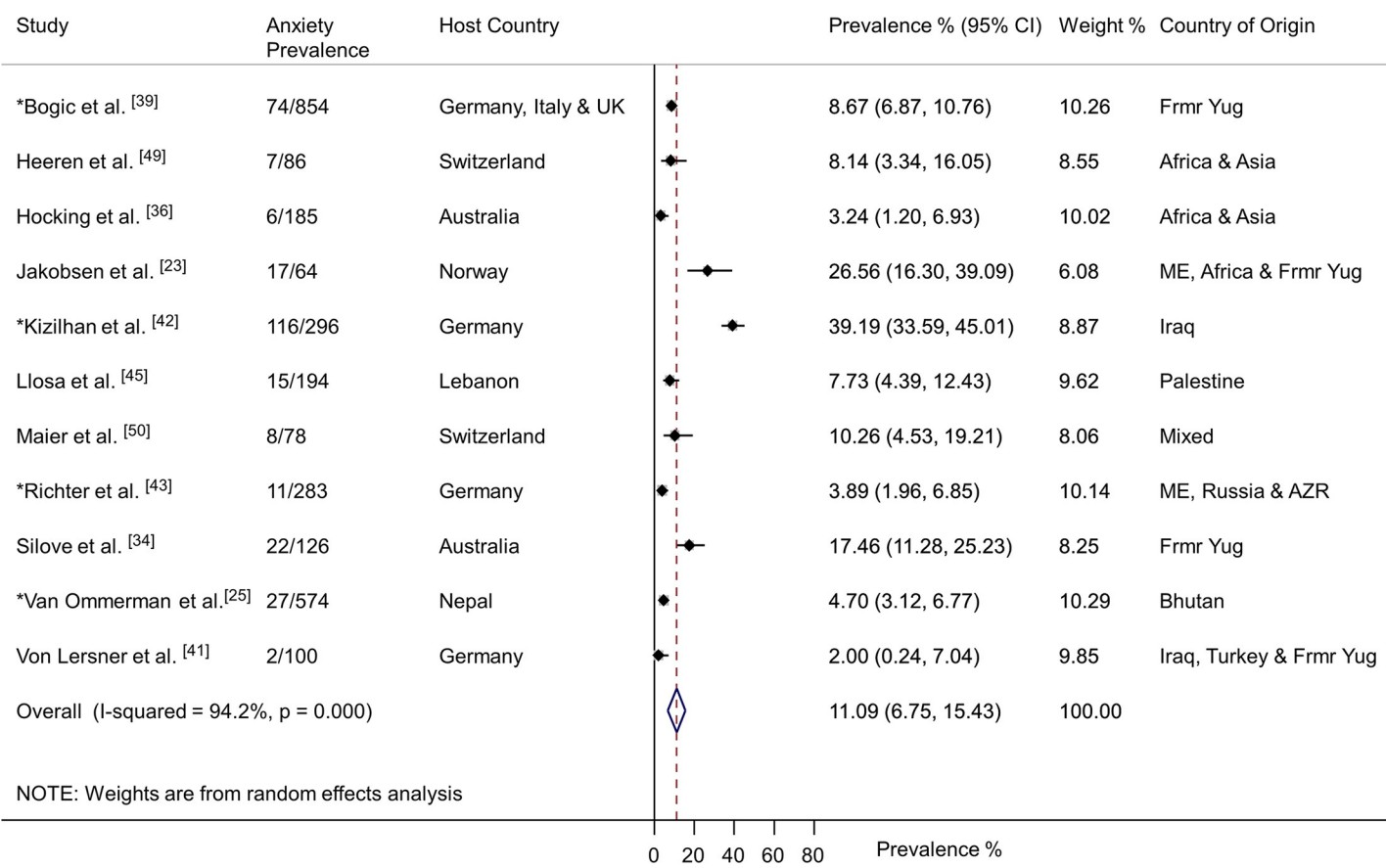

**Fig 6. Prevalence of anxiety in refugees and asylum seekers.** *Study with sample size of ≥200. Horizontal lines indicate 95% CIs; horizontal points of the open diamond are the limits of the overall 95% CIs; and the red dashed line shows the position of the overall prevalence. AZR, Azerbaijan; CI, confidence interval; Frmr Yug, former Yugoslavia; ME, Middle East.

psychological treatment for those affected by war [31,43]. One of these studies compared their help-seeking population to a randomly recruited sample, and there was a difference in prevalence rates, with higher rates in the help-seeking population [43].

## Discussion

Our results indicate that refugees and asylum seekers experience high rates of mental illness, in particular PTSD and depression. PTSD and depression appear to persist for many years post displacement, as there was no difference in prevalence between those displaced less than 4 years and those displaced longer. However, this was not the case for the prevalence of anxiety disorders, which we found to be higher among those displaced less than 4 years.

PTSD and depression in refugees and asylum seekers appear to be more prevalent than in the general population. According to data from the World Mental Health Surveys, lifetime prevalence in the general population is 3.9% for PTSD [57] and 12% for any depressive disorder [58], compared to our findings of 31% for PTSD and 31.5% for depression. However, the prevalence of anxiety disorders (11%) and psychosis (1.5%) in refugees and asylum seekers appears to be less than the lifetime prevalence in general population samples: 16% [58] and 3% [59], respectively. Only 11 studies reporting data on anxiety prevalence met the inclusion criteria for this review, and of those 11, only six assessed the full range of DSM anxiety disorders.

| Subgroup | Anxiety Prevalence | | Prevalence % (95% CI) | p-Value |
|---|---|---|---|---|
| **Size of Sample** | | | | |
| < 200 | 77/833 | | 9.24 (7.36, 11.42) | |
| ≥ 200 | 216/1995 | | 10.83 (9.50, 12.27) | 0.21 |
| **Duration Displaced** | | | | |
| < 4 years | 154/709 | | 21.72 (18.74, 24.94) | |
| ≥ 4 years | 64/942 | | 6.79 (5.27, 8.59) | <0.0001 |
| **Visa Status** | | | | |
| Asylum Seeker | 49/696 | | 7.04 (5.25, 9.20) | |
| Refugee | 244/2132 | | 11.44 (10.12, 12.87) | 0.0009 |
| **Region of Origin** | | | | |
| Asia | 27/574 | | 4.70 (3.12, 6.77) | |
| Africa | 3/30 | | 10.00 (2.11, 26.53) | |
| Middle East | 131/490 | | 26.73 (22.86, 30.89) | |
| Europe | 86/1068 | | 8.05 (6.49, 9.85) | |
| Mixed | 43/511 | | 8.41 (6.16, 11.17) | <0.0001 |
| **Residence** | | | | |
| Community | 107/1417 | | 7.55 (6.23, 9.05) | |
| Refugee Accommodation | 186/1411 | | 13.18 (11.46, 15.06) | <0.0001 |
| **Type of Interview** | | | | |
| Native Language | 243/2201 | | 11.04 (9.76, 12.40) | |
| Interpreter | 62/639 | | 9.70 (7.50, 12.30) | 0.34 |

Prevalence %

0    20    30    40

**Fig 7. Prevalence of anxiety by various study characteristics.** p-Values are derived from random-effects models; horizontal lines indicate 95% CI. Subgroup analysis for sex could not be conducted, owing to a lack of reported data. Subgroup analysis for diagnostic measure could not be conducted, owing to insufficient studies for each measure. CI, confidence interval.

With a heavy emphasis on PTSD and depression, the full breadth of anxiety disorders is less frequently examined and reported in the literature. It was only recently, with the release of DSM-5, that PTSD was no longer classified as an anxiety disorder but in a separate category of trauma and stressor-related disorders [60]. Further research on the prevalence of the full range of anxiety disorders and comorbidities is needed.

With the aim of including all possible refugee populations that have been studied, this systematic review placed few restrictions on characteristics of refugee experiences (region of origin or resettlement, duration of displacement, etc.). As a result, the review's criteria could in fact have been a contributing factor to the resulting substantial statistical heterogeneity. Despite this high heterogeneity, which is expected when investigating and analysing prevalence across global refugee populations, knowledge of current prevalence estimates provides a foundation for the field to build on. Researchers can progress with this knowledge and focus their attention on addressing the critical need for immediate, appropriate, and ongoing mental health support and interventions. Without the progression of further high-quality research that explores the different components of mental health needs, culturally appropriate and

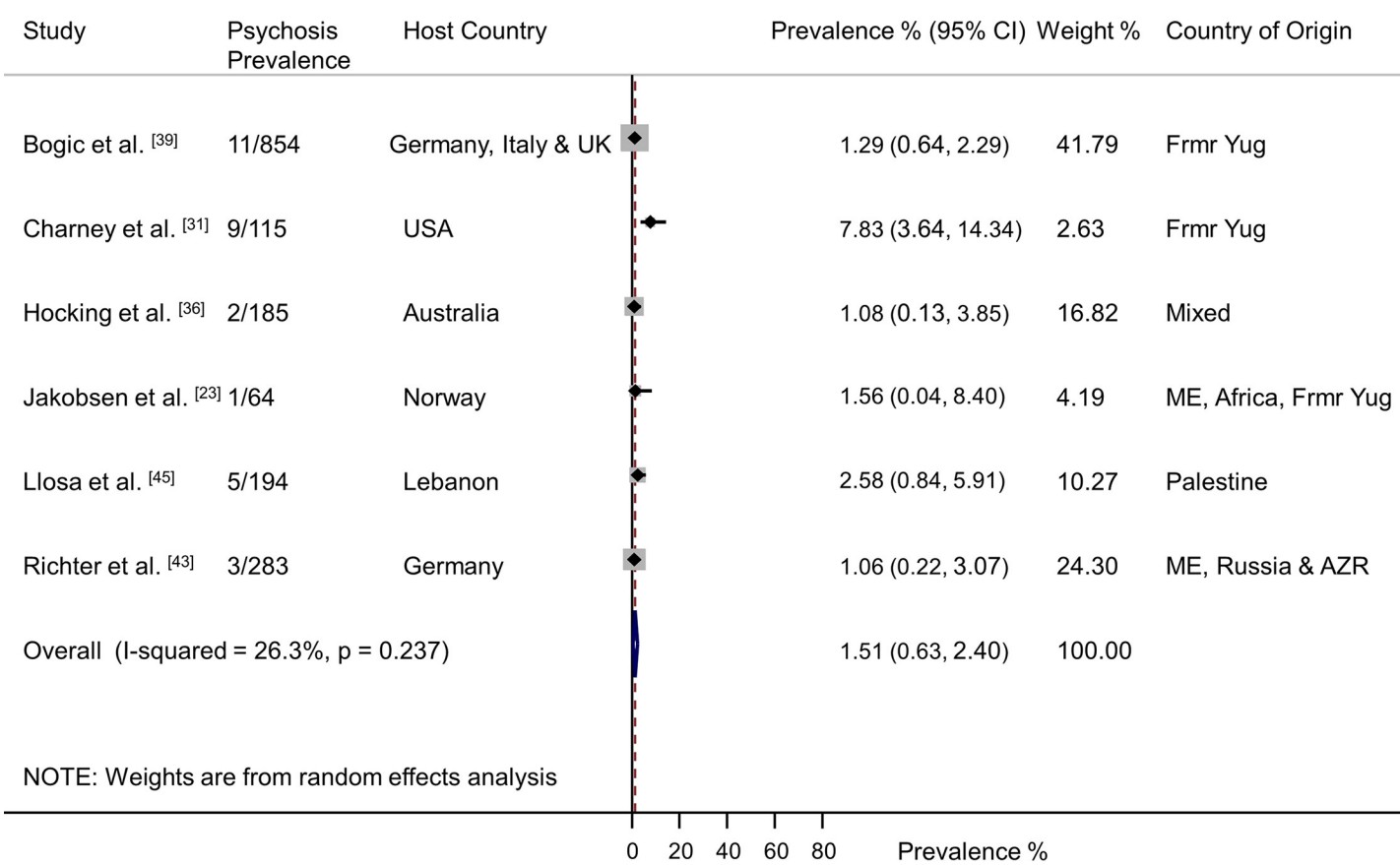

**Fig 8. Prevalence of psychosis in refugees and asylum seekers.** Horizontal lines indicate 95% CIs. Horizontal points of the open diamond are the limits of the overall 95% CIs; and the red dashed line shows the position of the overall prevalence. AZR, Azerbaijan; CI, confidence interval; Frmr Yug, former Yugoslavia; ME, Middle East.

effective interventions, and longitudinal mental illness trajectories, untreated mental illnesses will severely impact upon successful integration into host communities. For host countries and humanitarian agencies, current prevalence estimates of mental illness within this ever-growing population can be used in advocacy and health service planning to strengthen mental health services for refugees and asylum seekers, in line with WHO priorities and objectives [14].

Subgroup analysis for sex was only possible for PTSD, owing to a lack of sex data for the other outcomes, and this is a major limitation of the current literature. The subgroup analysis indicated a higher PTSD prevalence for women, consistent with studies of sex differences and PTSD within general populations [61–63]. During times of conflict, women face not only an increased risk of sexual violence [64–66], which is considered to confer a high risk for developing PTSD, but other risks associated with migration trauma such as safety concerns, child-rearing pressures, and exploitation and trafficking [67]. Although trauma type in relation to PTSD diagnosis was not adequately described in the studies, many of the studies included participants from countries such as the former Yugoslavia, Syria, and Iraq, areas with conflicts reported to have perpetrated systematic sexual violence [68]. In line with best-practice research reporting, future research in the field must ensure outcome measures are disaggregated by sex.

The studies with populations from Africa reported the highest prevalence of PTSD. This result likely reflects how countries within Africa are consistently ranked at the highest levels of the Political Terror Scale [69]. This scale is a five-point rating system based on data from

Amnesty International and the US State Department and measures the levels of extensive human rights violations and violence within nations. In our review, the refugee populations from Europe, which mostly consisted of individuals from the former Yugoslavia, had the highest prevalence of depression, and the Middle East refugee populations had the highest prevalence of anxiety.

The prevalence of PTSD and depression appeared to be higher in studies that utilised interpreter-assisted diagnostic assessments. However, this was not the case for anxiety disorders, for which we did not find evidence for a difference between the interpreter-assisted interviews and those conducted in the native language. This difference could be due to a number of factors, such as language fluency, which plays an important role in the diagnosis of mental illness because the clinician relies heavily on the self-reported symptoms of the individual [70]. However, further research is required to understand the differences in diagnosis rates between interpreter-assisted interviews and clinicians conducting the assessment in the native language and whether there are cultural and linguistic nuances that can impact on diagnostic rates that might only be accessible to native interviewers. Even though the different diagnostic measures are considered comparable in performance and diagnosis precision [71], our results suggest some differences, which highlight the importance of careful consideration of the method and instrument used in the mental health assessments of refugee populations. Although beyond the scope of this review, further investigation is required to understand potential differences in case identification between diagnostic measures.

Our findings suggest that the prevalence of PTSD and depression persists for many years post displacement, suggesting ongoing suffering from mental illnesses in the postmigration environment. This environment can include complexities of social and cultural isolation, reconfigured family relationships, difficulties adjusting to life in a foreign country, and often limited opportunities to contribute economically and socially to their new communities. Previous longitudinal studies have demonstrated how these hallmarks of the postmigration environment, alongside poor social support and acculturation difficulties, may contribute to a deterioration in mental health [5,72–74]. In contrast to the findings for PTSD and depression, anxiety prevalence was higher for those individuals recently displaced. Factors contributing to anxiety might be influenced by the uncertainty of the resettlement process and participation in the refugee determination process, which might have a detrimental effect on psychological well-being; however, robust longitudinal research is needed in this field.

We found that the prevalence of PTSD and depression is higher than in the review by Fazel and colleagues [6]. This could reflect the fact that this current systematic review included refugee populations from low- and middle-income countries or that the more recent refugee flows might be exposed to higher numbers of risk factors. In contrast, the results for anxiety disorders and psychosis are comparable with previously reported prevalence rates [6]. The influence of sample size is further supported, with the larger studies reporting lower prevalence rates for PTSD and depression. However, this was not the case for anxiety, for which sample size did not influence prevalence. The results for PTSD and depression are comparable to the findings by Steel and colleagues [7] and slightly lower than other systematic reviews, which have reported PTSD prevalence in the range of 36%–43% and depression 40%–44% [12,75].

Two phenomena currently affecting refugee and asylum-seeker populations should be considered when interpreting the results of this review. First is the increased targeting of civilian populations in areas of mass conflict. Second is the postmigration environment in countries with increasingly harsh immigration policies including detention, deportation, and delayed granting of refugee status—possibly mirroring local population shifts against immigration and heightened hostility towards refugee populations [76,77]. Investigation of these situations and their impact on mental health is warranted.

### Limitations and strengths

Some statistical heterogeneity is to be expected as a result of the review's design, which set no exclusion criteria for host country, country of origin, sex, or duration of displacement. We addressed this by using random-effects models to calculate more conservative 95% CIs. The conventional method to investigate potential sources of heterogeneity is to conduct a meta-regression; however, this was not possible, because of the limited covariates reported in the studies. We conducted subgroup analyses to investigate potential sources heterogeneity, but some subgroup analyses were also not possible, and some studies were excluded from sub-group analyses because of a lack of reported data. There are many challenges to conducting research with refugee populations, one of which is sampling. Ideally, this review would have restricted the inclusion criteria to studies that incorporated multistage representative sampling. However, such a restriction in this field would have yielded so few studies that the prevalence estimates could not have been made. In fact, only two of the included studies in this review would have met this criterion. Other limitations were imposed when studies combined illnesses to form diagnostic groups and/or reported only the number of comorbidities rather than the actual diagnoses. Although many of the diagnostic measures had been widely used in different cultural contexts, none had been specifically developed for refugee populations or cross-cultural use. Although the DSM-5 attempts to enhance cultural validity, all of the included studies used the DSM-IV, DSM-III-R, or ICD-10 criteria, previously criticized for limited recognition of cultural perspectives [78]. In particular, the diagnostic framework for PTSD has largely been investigated using military personnel and single-incident trauma survivors from high-income nations [79]. Somatic symptoms and related disorders were outside the scope of this review but warrant specific investigation and characterization.

As far as we are aware, this is the only systematic review to implement strict inclusion criteria regarding the diagnosis of mental illness in current refugee and asylum-seeker populations. This allowed for the selective analysis of higher-quality studies reporting the prevalence of mental illness based on clinical interviews with trained assessors using validated diagnostic measures. This review also expands the current evidence base by not only focusing on PTSD but also reporting depression, anxiety, and psychosis. To the best of our knowledge, this is the first systematic review to place no restrictions on language or on countries of origin or settlement. The majority of studies in this field are undertaken in high-income countries, which are often not countries of first asylum. Although most studies in this review came from countries such as the UK, Germany, Switzerland, and Australia, it also included studies from key refugee host nations such as Lebanon, Turkey, Uganda, and Nepal.

The ever-growing refugee and asylum-seeker populations pose a major global public health crisis with serious implications for mental health. This review provides current prevalence estimates for PTSD, depression, anxiety, and psychosis and suggests that both short-term and ongoing mental health services, beyond the period of initial resettlement, are required to promote the health of refugees.

## Supporting information

**S1 Prisma Checklist. From [16].** For more information, visit: www.prisma-statement.org.
(DOCX)

**S1 Table.** *Truncation symbol.** MeSH term, Medical Subject Headings.
(DOCX)

**S1 Risk of Bias.**
(DOCX)

**S1 Egger's Test PTSD.** Figure: Funnel plot using data from 22 studies providing data for the prevalence of posttraumatic stress disorder. Each dot represents a study. ES, effect size; s.e, standard error. Table: Egger's test set at a threshold of a *p*-value less than 0.05 to indicate funnel plot asymmetry. Coef., coefficient; Conf. Interval, confidence interval; Std_Eff, standard effect; Std. Err, standard error; Test of HO, test of null hypothesis.
(DOCX)

**S2 Egger's Test Depression.** Figure: Funnel plot using data from 17 studies providing data for the prevalence of depression. Each dot represents a study. ES, effect size; s.e, standard error. Table: Egger's test set at a threshold of a *p*-value less than 0.05 to indicate funnel plot asymmetry. Coef., coefficient; Conf. Interval, confidence interval; Std_Eff, standard effect; Std. Err, standard error; Test of HO, test of null hypothesis.
(DOCX)

**S3 Egger's Test Anxiety.** Figure: Funnel plot using data from 11 studies providing data for the prevalence of anxiety disorders. Each dot represents a study. ES, effect size; s.e, standard error. Table: Egger's test set at a threshold of a *p*-value less than 0.05 to indicate funnel plot asymmetry. Coef., coefficient; Conf. Interval, confidence interval; Std_Eff, standard effect; Std. Err, standard error; Test of HO, test of null hypothesis.
(DOCX)

**S4 Egger's Test Psychosis.** Figure: Funnel plot using data from six studies providing data for the prevalence of psychosis. Each dot represents a study. ES, effect size; s.e, standard error. Table: Egger's test set at a threshold of a *p*-value less than 0.05 to indicate funnel plot asymmetry. Coef., coefficient; Conf. Interval, confidence interval; Std_Eff, standard effect; Std. Err, standard error; Test of HO, test of null hypothesis.
(DOCX)

## Acknowledgments

We thank the following authors for providing additional information regarding their studies: C. Acarturk, M. Aoun, C. Eckart, E. Kaltenbach, C. J. Laban, A. Nickerson, F. Neuner, A. Rasmussen, Z. Steel, and S. Thapa. We would also like to thank A. Young from the Monash University library for her assistance with conducting the database search. We sincerely thank the Monash University staff and students and non-Monash colleagues who assisted with the screening of articles across a number of languages: R. Goldstein, C. Tay, C. Pickett (Edith Cowan University and Victoria University), D. Coles, K. Petersen, N. Pekin, K. Hammarberg, K. Stanzel, and R. Hasanov.

## Author Contributions

**Conceptualization:** Rebecca Blackmore, Jacqueline A. Boyle, Mina Fazel, Kylie M. Gray, Marie Misso, Melanie Gibson-Helm.

**Data curation:** Rebecca Blackmore, Grace Fitzgerald, Melanie Gibson-Helm.

**Formal analysis:** Rebecca Blackmore, Jacqueline A. Boyle, Sanjeeva Ranasinha.

**Investigation:** Rebecca Blackmore, Grace Fitzgerald, Melanie Gibson-Helm.

**Methodology:** Rebecca Blackmore, Jacqueline A. Boyle, Mina Fazel, Sanjeeva Ranasinha, Kylie M. Gray, Marie Misso, Melanie Gibson-Helm.

**Project administration:** Rebecca Blackmore.

**Supervision:** Jacqueline A. Boyle, Kylie M. Gray, Melanie Gibson-Helm.

**Visualization:** Rebecca Blackmore, Sanjeeva Ranasinha.

**Writing – original draft:** Rebecca Blackmore.

**Writing – review & editing:** Rebecca Blackmore, Jacqueline A. Boyle, Mina Fazel, Sanjeeva Ranasinha, Kylie M. Gray, Grace Fitzgerald, Marie Misso, Melanie Gibson-Helm.

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
