## [Decision Letter · Decision Letter 0]

4 Nov 2019

Dear Dr. Gibson-Helm,

Thank you very much for submitting your manuscript "The Prevalence of Mental Illness in Refugees and Asylum Seekers: a systematic review and meta-analysis" (PMEDICINE-D-19-02869) for consideration in PLOS Medicine's Special Issue on Refugee and Migrant Health. 

Your paper was evaluated by a senior editor and discussed among all the editors here. It was also discussed with the Guest Editors, and sent to three independent reviewers, including a statistical reviewer. The reviews are appended at the bottom of this email and any accompanying reviewer attachments can be seen via the link below:

[LINK]

In light of these reviews, I am afraid that we will not be able to accept the manuscript for publication in the journal in its current form, but we would like to consider a revised version that addresses the reviewers' and editors' comments. Obviously we cannot make any decision about publication until we have seen the revised manuscript and your response, and we plan to seek re-review by one or more of the reviewers. 

Due to the fact that you have a manuscript under consideration elsewhere describing findings related to those reported here, we ask that you submit your unpublished manuscript as supporting information and describe in your cover letter the differences between the two papers. The editorial team will discuss any potential overlap between the two reports, and consult with the Special Issue Guest Editors, and this will be taken into account when considering the revised manuscript.

We expect to receive your revised manuscript by Nov 25 2019 11:59PM. Please email us (plosmedicine@plos.org) if you have any questions or concerns.

We look forward to receiving your revised manuscript. 

Sincerely,

Caitlin Moyer, Ph.D.

Associate Editor 

PLOS Medicine

plosmedicine.org

1.Abstract: Please structure your abstract using the PLOS Medicine headings (Background, Methods and Findings, Conclusions). Please report your abstract according to PRISMA for abstracts, http://www.plosmedicine.org/article/info:doi/10.1371/journal.pmed.1001419 .

2. Abstract: Background (second sentence): Please revise to “...has reported wide variation in mental illness prevalence data....”

3. Abstract: Methods and Findings: Please provide the data sources, types of study designs included, eligibility criteria, and the synthesis and appraisal methods.

4. Abstract: Methods and Findings: In the last sentence of the Abstract Methods and Findings section, please describe the main limitation(s) of the study's methodology.

5. Author Summary: At this stage, we ask that you include a short, non-technical Author Summary of your research to make findings accessible to a wide audience that includes both scientists and non-scientists. The Author Summary should immediately follow the Abstract in your revised manuscript. This text is subject to editorial change and should be distinct from the scientific abstract. Please see our author guidelines for more information: https://journals.plos.org/plosmedicine/s/revising-your-manuscript#loc-author-summary

6. Methods: Please update your search to the present time. We require that SRs are updated to within roughly six months of the expected publication date. Your search has not been re-run since February of 2018.

7. Methods (Lines 144-145): Please describe the evaluation of study quality. Specifically, it is stated in the methods that: “Individual items related to study quality such as internal and external validity, reporting bias, and conflict of interest were assessed.” Please describe how these items were analyzed, and how this assessment factored into the results of the review and meta-analysis.

8. Methods (Lines 128-130): Please describe the random effects model used in the meta-analysis.

9. Methods: Please remove the section titled: “Role of the Funding Source” as this information is extracted from the manuscript submission system automatically.

10. Results: Please provide 95% CIs and p values for the results discussed for the subgroup analyses for each mental illness described (e.g. PTSD paragraph, depression paragraph, etc.).

11. Results: Please provide numerators and denominators for overall prevalence rates, if not in the text then at a minimum present these in the appropriate tables.

12. Discussion: Please expand on your Discussion as follows: Please increase the discussion of the existing research on prevalence of mental illnesses in the refugee and asylum seeking population, extending the depth of your discussion beyond the 2005 Fazel et al. and 2009 Steel et al. reviews if possible. Please expand on your discussion of subgroup analysis findings regarding why PTSD prevalence may be higher in individuals originating from Africa.

13. Discussion (Line 144-145, second to last paragraph): Please avoid assertions of primacy and add “To the best of our knowledge…” or similar to the following sentence: “It is also the first systematic review to place no restrictions on language or on countries of origin or settlement.”

14. Figures 2, 4, 6, 8, and 9: Please describe in the figure legend the meaning of the vertical dashed red line.

15. Figures 2-9: Please provide an X-axis label for these graphs.

16. References: Please use brackets for in-text reference numbers, e.g. [1].

17. Supporting Information: Please provide separate labels and titles (e.g. S1 Text, S1 Table, S1 Figure), and legends for all figures and tables. This includes the material included in your Appendix (Checklist, example search string, bias assessment template, and the three Egger plots for PTSD, Depression, and Anxiety). Please refer to our guidelines at:

https://journals.plos.org/plosmedicine/s/submission-guidelines#loc-supporting-information

18. Supporting information: Please define all abbreviations used within the figures and tables in the accompanying legends.

19. Supporting information: In your cover letter, you mentioned that the findings related to children and adolescents have been written up separately and that report is under consideration at another journal. Please include a copy of the unpublished manuscript as part of the supporting information with your revised submission, and also include a paragraph in your cover letter describing the key differences and any overlap between the two papers.

20. Checklist: Thank you for providing the PRISMA checklist. Instead of page numbers, please use sections and paragraphs when referring to locations within the article.

Comments from the reviewers:

Reviewer #1: Thank for the opportunity to review this paper. Blackmore and colleagues report findings from a systematic review which synthesizes mostly survey-based studies to describe the prevalence of mental illness in adult refugees and asylum seekers, reporting pooled results in post-traumatic stress disorder, depression and anxiety. All prevalence figures were higher in refugees than in the general population, which in itself not surprising. This paper adds updated figures that are more recent compared to some previous reviews done completed previously. I do, however, have some suggestions/comments primarily on the methodology. 

Abstract - in general the abstract needs a bit more detail when reporting the methods and findings, in particular

Lines 38-39: What "strict" inclusion criteria was described here? This is presented in the main text but the abstract should also explicitly state this directly 

Lines 40-41: Study quality assessed should be stated/assessed by the specific tool used. 

Abstract methods in general: Methods don't specify what type of meta-analyses was utilised? Random-effects and type of weighting method should be stated. Specify also this was combining aggregate prevalence measures reported in each study. 

Line 44: This jumps out without any context - meta-analyses performed in just larger studies, then this needs to pre-specified in the methods section of the abstract or it just looks like selective reporting of a particular sub-analyses.

Methods 

Firstly, my main concern on the methods is that as stated in Lines 93-96, the search strategy was not entirely the same as the earlier review performed by Fazel et al., and thus I question whether the rationale to limit the date to 1 Jan 2003 onwards appropriate. Because the search strategy is in fact different in this current review, it is not a "pure" an update on the Fazel et al review. The expanded search, with different criteria, pre-2003 may in fact pick up slightly different composition of studies from Fazel et al. because the authors imposed stricter criteria on mental health diagnoses and expanded the range of databases searched, number of search terms, and no restriction on geography or language. 

Second, even if the rationale was purely to update the Fazel et al. review, using the exact search strategy, any evidence synthesis and meta-analyses performed should also include the studies identified in the previous date for the update. Normally, Cochrane Review updates would in fact include both the previous studies combined with new studies. Here, the focus is only on new studies, but I'm not sure this is entirely rationale approach or at least has not been rationalised strongly enough. 

Line 103: Clarify if there were any restrictions on study design: i.e. cohort, case-control, cross-sectional surveys (though it looks like most studies were surveys)

Lines 129-130: What weighting methods was utilised in the random-effects model

Lines 134-135: This is the primary concern with the weighting methods. The authors state earlier they used a random-effects model to account for heterogeneity but then describe here that "prevalence rates were combined by direction summation of numerators and denominators". This would suggest that the prevalence rates were simply combined by direct summation across studies, which of course does not account for unequal weighting given from various studies due to sample size and heterogeneity in study design. The definition of prevalence implies a standard statistical assumption following a binomial distribution. Hence, the pooling of prevalence needs to consider the variance derived from the binomial distribution, accounting for the size of the study: var(p) = p(1-p)/N, p is the prevalence and N is the population size

Then the pooled prevalence can be combined using the inverse variance method and the model should be specified with a random-effects term to account for heterogeneity. 95% CI can be appropriately be computed using either the exact method, score method, Wald's method. see for the methodology: https://jech.bmj.com/content/67/11/974

It can be implemented in STATA using metaprop command: see https://archpublichealth.biomedcentral.com/articles/10.1186/2049-3258-72-39

Lines 138-139: Limiting studies stratified by participant number: This shouldn't be necessary with proper weighting methods when pooling (such as Inverse-variance or DerSimonian and Laird method), as small study effects will have large 95% CI and contribute fairly small effects to the overall pooled results. Doing this arbitrary stratification in the primary analyses actually introduces some bias itself. 

Lines 142-143: The NOS scale is useful but really designed for Cohort and Case-control studies, hence most of the questions refer to selection bias, control for confounding, and selection of comparators, which all cross-sectional studies are not designed to capture. AXIS-tool is more relevant for assessing quality of cross-sectional studies https://bmjopen.bmj.com/content/6/12/e011458

Overall impression: The study does has it's merits and appreciate the authors hard work in an important area which has a strong rationale to undertake this research. However, there are some aspects the authors should address and clarify. In my opinion, the rationale for limiting the start date with an altered set of search criteria from a previous review is not strong enough. This would have been far more comprehensive as a review to also include studies pre-2003. In fact, there was an opportunity to present a time-stratified sub-analyses to look if the prevalence of mental illness have increased or decreased over time. There also some questions I have on the method which the prevalence figures were synthesized, including how the prevalence figures were pooled - so it's a bit difficult interpret how robust the results are without these factors in clarified and considered.

Reviewer #2: Thank you for your work on this systematic review and meta-analysis on the Prevalence of Mental Illness in Refugees and Asylum Seekers. 

Strengths of your manuscript include:

- expanding the evidence base on psychosis among refugees and asylum seekers

- This work continues to demonstrate that refugees experience long term psychiatric sequelae of traumatic events experienced as a result of refugee status and as you note call for long-term health care beyond the initial period of resettlement- very important to ensure funding and programmatic planning.

- Very important finding on:"We found that PTSD and depression were higher for those displaced longer than four years, suggesting a possible deterioration of mental health in the post-migration environment".

I am suggesting several points that should be addressed prior to publication that can improve the utility of this work:

- My main concern is that although it is noted that there is some novel information in this review, besides the interesting information on psychotic symptoms, this review essentially replicates what we know from the literature on PTSD for the past 20 years. The problem is a political one that translates to underfunded programs and poor to no policies based on the evidence base, globally. The majority of refugees and asylum seekers do not live in high income countries- they face significant challenges in settings where there is little to no mental health care. As such, it would be good if you could address, even very briefly, the broader context for having up to date estimates of poor mental health among refugees and asylum seekers globally. Chronic PTSD prevents integration into new societies and reintegration. Funding such programs now makes good economic sense as well as having a human rights imperative. 

- It is discussed that it's a strength that results from medical settings were included (excluding survey results) but I'm not sure that's a strength. While in some settings, it can be "assumed" that this means that a diagnosis of say, PTSD, is likely more accurate, it still depends on the clinician, their level of training, need for a quick diagnosis etc etc. It also has a potential country setting bias- biased towards places that may have higher capacity to diagnose and treat PTSD. This should be clearly stated. 

- It's concerning that there is such heterogeneity among study results, even though the authors admirably sought to address this. It could be due to your review's criteria and that the results are drawn from such different settings and populations- these populations have experienced such different experiences and culturally may manage them differently. THe authors state that this means that "The results of the meta-analysis yielded high statistical heterogeneity, which is evidence of the critical need for research in this field that is large-scale, uses rigorous diagnostic methods, and characterizes the study sample in detail." but this is highly unlikely as the reason, given the 20 years of replicable results already clear in the literature. Again, this is likely the result of the review's criteria. 

- The lower anxiety results is puzzling. PTSD and anxiety disorders are highly comorbid. More explanation needs to be given for this including whether the review's strategy led to potentially inaccurate results.

Given all of the above, the review needs major revisions but could still be a helpful publication if strengthened. 

Reviewer #3: The authors have undertaken a systematic review and meta-analysis of epidemiological studies reporting the prevalence of mental disorder amongst refugee and asylum seekers. The review reported is very closely modelled on an earlier review undertaken by Fazel et al (2005). As with that review the authors have restricted their review to research studies that have used structured or semi-structured diagnostic instruments and have excluded studies that report prevalence estimates derived from screening or self-report measures. 

The prevalence estimates cited on page 3, line 72-74 are not so relevant to the current review given the inclusion of post-conflict country surveys in that review. It may be possible to cite more specific displaced population estimates. 

At line 101, page 4 the authors should clarify the number of systematic reviews examined and include citations for these.

A difference between the Fazel et al study and this research which should be highlighted or corrected is that the authors have included LMI countries of first asylum studies of refugees and asylum seekers. The Fazel et al study was restricted to resettled refugees or asylum seekers in Western or HIC country settings. 

It is not clear that the search criteria are optimised to identify those displaced with the region as other terms are often applied. There are a number of studies amongst displaced populations that appear to have used structured diagnostic measures that may be relevant to stated inclusion criteria. I have included a list of studies that the authors should consider. If some of these do meet inclusion criteria then there may be problems with the search strategy applied.

It is not clear what data was extracted by the authors - the manuscript lists sample size, publication year, and country or region of origin. Meta-analytic stratification suggest that other data was extracted such as, sex specific prevalence rates; duration of displacement, and living circumstance. Additional information should be provided on this and whether data on sex especially was extracted as a percentage distribution of extracted separately for makes and females, the later being preferable. 

It is a shame that information on the prior trauma and torture exposure given the importance of these as determinants of MH outcomes, although noted that this was not undertaken by Fazel but has been undertaken by subsequent reviews. 

Page 28, I agree a strength to limit to diagnostic measures, probably should also limit to multi-stage representative sampling. The study by Fazel especially suggested that study with a sample of over 200 may be the lowest number to reach stable population estimates. 

Banal, R., J. Thappa, et al. Psychiatric morbidity in adult Kashmiri migrants living in a migrant camp at Jammu.Indian J Psychiatry 52: 154-158; 2010). 

Amowitz LL, Heisler M, Iacopino V. A population-based assessment of women's mental health and attitudes toward women's human rights in Afghanistan. Journal of Women's Health. 2003;12(6):577-587.

Eytan A, Durieux-Paillard S, Whitaker-Clinch B, Loutan L, Bovier PA. Transcultural validity of a structured diagnostic interview to screen for major depression and posttraumatic stress disorder among refugees. Journal of Nervous & Mental Disease. 2007;195(9):723-728.

Fenta H, Hyman I, Noh S. Determinants of depression among Ethiopian immigrants and refugees in Toronto. Journal of Nervous & Mental Disease. 2004;192(5):363-372.

Marshall GN, Schell TL, Elliott MN, Berthold SM, Chun C-A. Mental health of Cambodian refugees 2 decades after resettlement in the United States. JAMA. 2005;294(5):571-579.

Renner W, Salem I, Ottomeyer K. Cross-cultural validation of measures of traumatic symptoms in groups of asylum seekers from Chechnya, Afghanistan, and West Africa. Social Behavior and Personality. 2006;34(9):1101-1114. 

Renner W, Salem I. Post-traumatic stress in asylum seekers and refugees from Chechnya, Afghanistan, and West Africa: gender differences in symptomatology and coping. International Journal of Social Psychiatry. 2009;55(2):99-108.

Toscani L, Deroo LA, Eytan A, Gex-Fabry M, Avramovski V, Loutan L, Bovier P. Health status of returnees to Kosovo: do living conditions during asylum make a difference? Public Health. 2007;121(1):34-44.

[LINK]

---

## [Decision Letter · Decision Letter 1]

17 Jun 2020

Dear Dr. Gibson-Helm,

Thank you very much for submitting your revised manuscript "The Prevalence of Mental Illness in Refugees and Asylum Seekers: a systematic review and meta-analysis" (PMEDICINE-D-19-02869R1) for consideration in PLOS Medicine's Special Issue on Refugee and Migrant Health.

I apologize for the delay in review. Your paper was evaluated by a senior editor and discussed among all the editors here. It was also discussed with the Special Issue Guest Editors, and was re-reviewed by the statistical reviewer. The reviews are appended at the bottom of this email and any accompanying reviewer attachments can be seen via the link below:

[LINK]

In light of these reviews, I am afraid that we will not be able to accept the manuscript for publication in the journal in its current form, but we would like to consider a revised version that addresses the reviewers' and editors' comments. Mainly, we request that you please be sure to address the first comment of Reviewer 1, regarding the analysis comparing studies with native language vs. interpreter-assisted assessments. There were also a few points raised during the previous round of reviews that we would like to see further clarified (please see the Editor's list of requests below).

Obviously we cannot make any decision about publication until we have seen the revised manuscript and your response, and we plan to seek re-review by one or more of the reviewers. 

We expect to receive your revised manuscript by Jun 24 2020 11:59PM. Please email us (plosmedicine@plos.org) if you have any questions or concerns.

We look forward to receiving your revised manuscript. 

Sincerely,

Caitlin Moyer, Ph.D.

Associate Editor 

PLOS Medicine

plosmedicine.org

1. Reviewer 1, point #1: Please do address this point raised by the reviewer, perhaps conducting a quick analysis to determine if there was any difference between native language and interpreter-assistance studies. 

2. Response to Reviewers: Reviewer 2, point #2: In response to reviewer 2’s second point, I think please explicitly clarify in the text that you have included studies that recruited individuals from general refugee health/ primary health care clinics.

3. Response to Reviewers: Reviewer 2, point # 3: Please discuss ways in which the review's criteria could be a potential contributing factor to the heterogeneity observed.

4. Response to Reviewers: Reviewer 3, point #1: Please more clearly discuss that a limitation of some of the cultural/nation specific prevalence estimates (such as those that were cited) is that some estimates focus on internally displaced/conflict-affected populations and a global refugee comprehensive review is needed.

5. Response to Reviewers: Reviewer 3: final point: Please discuss that restricting inclusion criteria to studies which used multi-stage representative sampling is a limitation (perhaps discuss within the discussion section), and please report in the manuscript how many of their studies would have been excluded based on this restriction.

6. Abstract: Line 48-49: Please clarify this sentence: “Random effects, based on inverse variance weights, were conducted.” Do you mean “random effects models” were conducted?

7. Abstract: Please make some mention of the countries, or at least number of countries, represented by the review, such as described in your results (or similar): “Studies were undertaken in 15 countries with participants from four geographical regions: the Middle East (43%), Europe (29%), Asia (20%), Africa (5%).”

8. Abstract: Methods and Findings: Please revise this sentence to clarify as follows: “The prevalence of post traumatic stress disorder (PTSD) was 31.46% (95% CI 24.43-38.5), the prevalence of depression was 31.5% (95% CI 22.64-40.38), the prevalence of anxiety disorders was 11% (95% CI 6.75-15.43), and the prevalence of psychosis was 1.51% (95% CI 0·63-2·40).” or similar.

9. Abstract: Limitations statement: Thank you for including a statement of limitations, please revise to: “A limitation of the study is that substantial heterogeneity was present in the prevalence estimates of PTSD, depression, and anxiety, and limited covariates were reported in the included studies.”

10. Abstract: Line 59: Please remove the subjective term “rigorous”

11. Author summary: What do these findings mean? Line 77: Should this read “increased prevalence” in the first bullet point?

12. Introduction: Line 94, and throughout manuscript where applicable: Please refer to low or middle income countries rather than "developing countries" or "the Global South". Please refer to high income countries rather than "developed" or "Western" countries. 

13. Results Line 232-233: Please also provide 95% CIs for duration of displacement results.

14. Results Line 247-249: Please clarify “significantly lower” than which comparison group in the following sentence: “In the seven larger studies with 200 or more participants, depression prevalence was significantly lower, 20.65% (95% CI 18.88-22.51).” If statistical significance is meant, please include associated p values in addition to CIs.

15. Results Line 249-250: Please also provide 95% CIs for duration of displacement results.

16. Results Line 271: Please also provide 95% CIs for the sample size and anxiety disorder analysis.

17. Results Line 295-297: Please clarify that you meant “resulting in” and not “...and this was associated with fewer than expected…” in the following sentence: “Additionally, in one study only male psychologists conducted the diagnostic assessments, resulting in fewer than expected reports of sexual assault. [51]”

18. Discussion: Line 314: Please indicate that you mean “anxiety prevalence” here.

19. Discussion: Line 344-346: Please revise this sentence to clarify: Although trauma type in relation to PTSD diagnosis was not adequately described in the studies, many studies reported on participants from countries such as the former Yugoslavia, Syria, and Iraq, areas with conflicts reported to have perpetrated systematic sexual violence.” or similar depending on your intended meaning here.

20. Discussion: Line 353: Please revise this sentence, without overreaching what can be concluded from the data- we suggest: “Our findings suggest that the prevalence of PTSD and depression persists for many years post-displacement…” Similarly at line 363, we suggest beginning this sentence with “We found that…”

21. References: In-text citations should be in regular text (not superscript), with numbers in brackets appearing before punctuation, like this: [1].

22. Reference List: Please use the "Vancouver" style for reference formatting, and see our website for other reference guidelines: https://journals.plos.org/plosmedicine/s/submission-guidelines#loc-references

23. Supporting information: S1 Table, S3-S6 Eggers test results: Please provide titles and descriptive legends for each individual table and figure in the Supporting Information.

Comments from the reviewers:

Reviewer #1: The revised version of the manuscript is much approved by the authors. The authors have spent significant effort addressing review amendments and suggestions. The responses were appropriate and the corresponding manuscript has reflected the suggested changes. 

Two final additional points that I felt could potentially enhance this nice piece of work to consider:

1) There were thirteen studies conducting assessment in native language compared to thirteen with assistant from interpreters. I wonder if there was any influence on prevalence figures - which would have implications on the mode of assessment in native languages. 

2) The authors picked up five diagnosis measures - were there enough studies in any one of those diagnostic measures to conduct a sub-group that were diagnostic measure specific. Again - if possible the results could indicate what particular measures may be contributing to the heterogeneity and whether there is any indication that certain diagnostic measures may be more sensitive towards picking up higher levels of the outcomes of interest. 

Apart from these comments - I think the manuscript is publishable.

[LINK]

---

## [Decision Letter · Decision Letter 2]

23 Jul 2020

Dear Dr. Gibson-Helm,

Thank you very much for re-submitting your manuscript "The Prevalence of Mental Illness in Refugees and Asylum Seekers: a systematic review and meta-analysis" (PMEDICINE-D-19-02869R2) for review by PLOS Medicine.

I have discussed the paper with my colleagues and the academic editor and it was also seen again by one of the reviewers. I am pleased to say that provided the remaining editorial and production issues are dealt with we are planning to accept the paper for publication in the journal.

[LINK]

We look forward to receiving the revised manuscript by Jul 30 2020 11:59PM. 

Sincerely,

Caitlin Moyer, Ph.D.

Associate Editor 

PLOS Medicine

plosmedicine.org

Requests from Editors:

1.Results Line 289-290: Please provide 95% CIs to accompany the p value reported here: “The use of an interpreter to conduct assessments had no significant impact on the reported prevalence of anxiety disorders (p = 0.34).”

2.Discussion: first paragraph, lines 334-335: Please revise this sentence: “However, this was not the case for the prevalence of anxiety disorders, which we found to be higher among those displaced less than four years.” or similar, to clarify.”

3.Discussion: Line 340: Should this read “...less than the lifetime prevalence…”?

4.Discussion: Line 377: Please revise: “anxiety disorders, where there was no difference between…” to read “...anxiety disorders, where we did not find evidence for a difference between…”

5.Discussion: Line 385-386: Please revise to clarify this sentence, we suggest: “Although beyond the scope of this review, further investigation is required to understand potential differences in case identification between diagnostic measures.” or similar.

6.Discussion: line 399-400: Please revise to clarify this sentence. We suggest: “In contrast, the results for anxiety disorders and psychosis are comparable with previously reported prevalence rates [6].” or similar.

7.Discussion: Line 416: Please change to “investigate potential sources of heterogeneity”

8.Figure 2: Please define the abbreviations for PTSD and CI in the legend.

9.Figure 3: Please provide an X axis label. Please spell out abbreviations for PTSD, CI, and for the diagnostic measures MINI, CAPS, WHO-CIDI, and SCID in the legend.

10.Figure 4: Please define the abbreviation for CI in the legend.

11.Figure 5: Please spell out the abbreviations for CI, MINI, WHO-CIDI, and SCID. Please provide an X axis label.

12.Figure 6: Please spell out the abbreviation for CI in the legend.

13.Figure 7: Please spell out the abbreviation for CI in the legend, and please provide an X axis label.

14.Figure 8: Please spell out the abbreviation for CI in the legend.

15. Supporting information files S3-S6: Please provide legends for each file in which you describe all figure panels and spell out all abbreviations used within the figures.

Comments from Reviewers:

Reviewer #1: I agree that the authors have done their part in adding the additional analyses. How interesting the results!

This implication being I think being that when designing these assessments - the method of assessment and diagnostic measure had a significant bearing on the proportion of cases being detected (the direction of association seemed to be consistently one way as well).

The limitation is that we don't of course know what is closer to the true answer without a known reference standard (arguably in this population this would be difficult to achieve) but it does raise the issues that careful consideration of the method of assessment and instrument of assessment in this population.

[LINK]

---

## [Editor Report · Decision Letter 3]

14 Aug 2020

Dear Dr Gibson-Helm, 

On behalf of my colleagues and the academic editor, Dr. Paul Spiegel, I am delighted to inform you that your manuscript entitled "The Prevalence of Mental Illness in Refugees and Asylum Seekers: a systematic review and meta-analysis" (PMEDICINE-D-19-02869R3) has been accepted for publication in PLOS Medicine. 

PRODUCTION PROCESS

PRESS

PROFILE INFORMATION

Thank you again for submitting the manuscript to PLOS Medicine. We look forward to publishing it. 

Best wishes, 

Caitlin Moyer, Ph.D.

Associate Editor 

PLOS Medicine

plosmedicine.org